# Decoding odor quality and intensity in the *Drosophila* brain

Antonia Strutz[1], Jan Soelter[2], Amelie Baschwitz[1], Abu Farhan[1], Veit Grabe[1], Jürgen Rybak[1], Markus Knaden[1], Michael Schmuker[2†], Bill S Hansson[1], Silke Sachse[1]*

[1]Department of Evolutionary Neuroethology, Max Planck Institute for Chemical Ecology, Jena, Germany; [2]Department for Biology, Pharmacy and Chemistry, Free University Berlin, Neuroinformatics and Theoretical Neuroscience, Berlin, Germany

**Abstract** To internally reflect the sensory environment, animals create neural maps encoding the external stimulus space. From that primary neural code relevant information has to be extracted for accurate navigation. We analyzed how different odor features such as hedonic valence and intensity are functionally integrated in the lateral horn (LH) of the vinegar fly, *Drosophila melanogaster*. We characterized an olfactory-processing pathway, comprised of inhibitory projection neurons (iPNs) that target the LH exclusively, at morphological, functional and behavioral levels. We demonstrate that iPNs are subdivided into two morphological groups encoding positive hedonic valence or intensity information and conveying these features into separate domains in the LH. Silencing iPNs severely diminished flies' attraction behavior. Moreover, functional imaging disclosed a LH region tuned to repulsive odors comprised exclusively of third-order neurons. We provide evidence for a feature-based map in the LH, and elucidate its role as the center for integrating behaviorally relevant olfactory information.

**\*For correspondence:** ssachse@ ice.mpg.de

**Present address:** †School of Engineering and Informatics, University of Sussex, Brighton, United Kingdom

**Reviewing editor**: Mani Ramaswami, Trinity College Dublin, Ireland

## Introduction

To navigate the environment in a way that optimizes their survival and reproduction, animals have evolved sensory systems. These have three essential tasks: First, the external world has to be translated into an internal representation in the form of an accurate neural map. Second, the neural map has to be readable and interpretable, that is, the generated neural code must allow common attributes to be extracted across stimuli to enable the animal to make the best decisions. Third, the animal has to be able to adapt to environmental changes and to form a sensory memory of new stimuli. Many studies have been dedicated to unraveling the primary transformation from a stimulus into an initial neural representation within various sensory systems (*Manni and Petrosini, 2004*; *Vosshall and Stocker, 2007*; *Sanes and Zipursky, 2010*) and to elucidating neuronal plasticity and sensory memory formation in higher-level processing centers (*Heisenberg, 2003*; *Pasternak and Greenlee, 2005*). The ability to extract features and integrate stimulus modalities have so far mainly been studied in the visual system (*Livingstone and Hubel, 1988*; *Bausenwein et al., 1992*; *Nassi and Callaway, 2009*). We addressed the question of how stimulus features such as odor valence and intensity are coded and integrated within the olfactory system using the model organism *Drosophila melanogaster*.

The olfactory system of the vinegar fly provides an excellent model system for deciphering olfactory processing mechanisms, since it displays remarkable similarities to the mammalian system but is less complex and highly genetically tractable. Like other sensory systems, the olfactory system employs a spatio-temporal map to translate the variables in chemosensory space into neuronal activity patterns in the brain. This map emerges when the olfactory sensory neurons (OSNs) with the same chemosensory receptors converge into one exclusive glomerulus in the antennal lobe (AL) which represents the

**eLife digest** Organisms need to sense and adapt to their environment in order to survive. Senses such as vision and smell allow an organism to absorb information about the external environment and translate it into a meaningful internal image. This internal image helps the organism to remember incidents and act accordingly when they encounter similar situations again. A typical example is when organisms are repeatedly attracted to odors that are essential for survival, such as food and pheromones, and are repulsed by odors that threaten survival.

Strutz et al. addressed how attractiveness or repulsiveness of a smell, and also the strength of a smell, are processed by a part of the olfactory system called the lateral horn in fruit flies. This involved mapping the neuronal patterns that were generated in the lateral horn when a fly was exposed to particular odors.

Strutz et al. found that a subset of neurons called inhibitory projection neurons processes information about whether the odor is attractive or repulsive, and that a second subset of these neurons process information about the intensity of the odor. Other insects, such as honey bees and hawk moths, have olfactory systems with a similar architecture and might also employ a similar spatial approach to encode information regarding the intensity and identity of odors. Locusts, on the other hand, employ a temporal approach to encoding information about odors.

The work of Strutz et al. shows that certain qualities of odors are contained in a spatial map in a specific brain region of the fly. This opens up the question of how the information in this spatial map influences decisions made by the fly.

equivalent to the mammalian olfactory bulb (*Hildebrand and Shepherd, 1997*; *Vosshall et al., 2000*; *Vosshall and Stocker, 2007*). Glomeruli, the functional and morphological units of the AL, are microcircuits comprising OSNs, multiglomerular local interneurons (LNs) and uniglomerular output neurons, so-called excitatory projection neurons (ePNs) (*Wilson and Mainen, 2006*; *Vosshall and Stocker, 2007*) that convey the olfactory information to higher brain centers, as the mushroom body calyx (MBc) and the lateral horn (LH) (*Stocker et al., 1997*). The stringent spatial arrangement of OSNs and ePNs in the AL generates a spatial map containing characteristic combinatorial glomerular activity patterns for all odorants (*Fiala et al., 2002*; *Wang et al., 2003a*; *Couto et al., 2005*; *Fishilevich and Vosshall, 2005*). The MBc is involved in olfactory memory formation (*Heisenberg, 2003*) and enables a contextualization of the odor space (*Caron et al., 2013*). By exclusion, the LH is believed to be involved in innate olfactory behavior (*de Belle and Heisenberg, 1994*; *Jefferis et al., 2007*). Excitatory PNs retain the sensory information encoded in the AL and form glomerulus-dependent, stereotypic axonal terminal fields in the LH (*Marin et al., 2002*; *Wong et al., 2002*; *Tanaka et al., 2004*). Compartmentalization in the LH has been observed in form of a spatial segregation of ePNs innervating specific glomerular subgroups (*Tanaka et al., 2004*), fruit and pheromone odor information processing ePNs (*Jefferis et al., 2007*) as well as ammonia and amine vs carbon dioxide coding ePNs (*Min et al., 2013*).

Like many other sensory networks, the olfactory circuit of the fly contains spatially distinct pathways to the higher brain, namely the inner, middle and outer antennocerebral tract (iACT, mACT and oACT) (*Stocker et al., 1990*). Notably, the mACT projects from the AL to the LH exclusively and consists of inhibitory PNs (iPNs), which exhibit also uniglomerular but mainly multiglomerular AL innervations (*Ito et al., 1997*; *Jefferis et al., 2007*; *Lai et al., 2008*; *Okada et al., 2009*; *Liang et al., 2013*). Both PN populations have been attributed different coding properties: Although both PN populations exhibit odor responses to overlapping odor ligands, iPNs seems to be broader tuned than ePNs (*Wang et al., 2014*). Furthermore, while ePNs encode rather odor identity (*Wang et al., 2003a*; *Wilson et al., 2004*; *Silbering et al., 2008*), iPNs have been shown to enhance innate discrimination of closely related odors (*Parnas et al., 2013*). Together, these PN populations process information on dual olfactory pathways (*Liang et al., 2013*; *Wang et al., 2014*), as do processing mechanisms in other sensory modalities (*Nassi and Callaway, 2009*), and most likely accomplish different olfactory behaviors. The mainly multiglomerular AL pattern of iPNs suggests that these neurons extract characteristic stimulus features from the AL code and re-integrate this information into the LH to mediate innate odorant-guided behavior. This assumption is further supported by two recent studies showing that the

inhibitory input from the AL to the LH is module-specific, that is, selective for food odors and phero-mones (*Liang et al., 2013*; *Fisek and Wilson, 2014*), while the connectivity in the MBc is rather prob-abilistic (*Murthy et al., 2008*; *Caron et al., 2013*).

However, it still remains open if and how different odor features as hedonic valence or intensity are functionally coded and integrated in the LH. In this study, we characterized and dissected the iPN olfactory processing pathway regarding the coding of odor quality and intensity at morphological, functional and behavioral levels. By linking odor-evoked activity patterns in the LH to odor-guided behavior, we provide evidence that iPNs mediate odor attraction. Furthermore, our data demonstrate a feature-based, spatially segregated activity map in the LH comprised of iPNs and third-order neu-rons and thus expand its role as a center for integrating behaviorally relevant olfactory information.

## Results

### Dendrites of iPNs innervate two-thirds of olfactory glomeruli

Cell bodies of iPNs are exclusively located in the ventral cell cluster which consists of ~50 iPNs (*Lai et al., 2008*) that project via the mACT to the LH, thereby bypassing the MBc (*Ito et al., 1997*) (*Figure 1A,B*). In contrast, ePN somata are located anterodorsally and laterally of the AL, and their axons project through the iACT or oACT to the MBc and the LH (*Stocker et al., 1997*; *Marin et al., 2002*; *Wong et al., 2002*; *Lai et al., 2008*). To analyze the innervation patterns of iPNs and ePNs, we labeled both PN populations simultaneously in vivo using the enhancer trap lines *GH146-QF* and *MZ699-GAL4* that label the majority of ePNs (60%) and iPNs (86%), respectively (*Lai et al., 2008*). Double-labeling shows that both PN types innervate overlapping regions in the AL and the LH, while a small posterior-lateral LH area is targeted only by ePNs (*Figure 1A*, *Figure 1—figure supplement 1*). In GH146-positive (GH146+) PNs, immunolabeling reveals GABA production in all ~6 PNs of the ven-tral cell cluster (*Wilson and Laurent, 2005*), whereas ePNs of this line are exclusively cholinergic (*Shang et al., 2007*). For the ~45 MZ699-positive (MZ699+) iPNs (*Lai et al., 2008*), GAD1 (glutamic acid decarboxylase) in situ hybridizations imply GABA synthesis (*Okada et al., 2009*), which was recently verified via immunostaining (*Liang et al., 2013*; *Parnas et al., 2013*). The polarity of both PN populations has been studied in detail, showing that both possess dendritic regions in the AL, indicating the AL as their cholinergic input site, while the LH represents their major output site (*Jefferis et al., 2001*; *Okada et al., 2009*; *Liang et al., 2013*; *Parnas et al., 2013*).

To further characterize PNs labeled by *MZ699-GAL4* and *GH146-GAL4*, we analyzed their precise glomerular innervation to unravel how selectively they acquire information in the AL. To allow glomer-ulus identification in vivo, we employed a transgenic fly carrying elav-n-synaptobrevin:DsRed (END1-2) to express the presynaptically targeted fusion protein under the control of the neuron-specific elav promotor (*Figure 1—figure supplement 2A*) (*Grabe et al., 2014*). The reconstruction and identifica-tion of all AL glomeruli provided 53 glomeruli, of which 75% were innervated by MZ699+ iPNs (40) while 70% (37) were covered by GH146+ ePNs (*Figure 1C*, *Figure 1—figure supplement 2B*). 55% of all glomeruli were innervated by both lines. Notably, dendritic *MZ699-GAL4* innervation density was not homogeneous. Certain glomeruli were densely innervated (e.g., DM2, DM4 and DM5), while others did not reveal any postsynaptic sites (e.g., DL1, DL4 and DL5). Hence MZ699+ iPNs target specific glomerular subsets selectively, which suggests that these neurons have a particular function within the olfactory network.

### Calcium signals in the lateral horn spatially segregate into distinct response domains

Probabilistic synaptic density maps of GH146+ PNs predicted a regionalized neuronal activity in the LH (*Jefferis et al., 2007*). Do iPNs functionally segregate in a comparable way? To address this ques-tion, we expressed the $Ca^{2+}$-sensitive reporter G-CaMP3.0 (*Nakai et al., 2001*; *Tian et al., 2009*) in iPNs using *MZ699-GAL4* and performed functional imaging in the LH (*Figure 2A–C*). We initially tested three odors with potential relevance for *Drosophila* at different concentrations: acetoin acetate, an attractive byproduct of the yeast fermentation process, balsamic vinegar, an attractive natural odor mixture, and benzaldehyde, a well-known fly repellant (*Magee and Kosaric, 1987*; *Keene et al., 2004*; *Semmelhack and Wang, 2009*). We observed that odor evoked $Ca^{2+}$ responses separate in certain regions of the LH in an odor-specific and concentration-dependent manner (*Figure 2C*). Acetoin ace-tate and balsamic vinegar evoked $Ca^{2+}$ activity in spatially similar regions. At higher concentrations, an

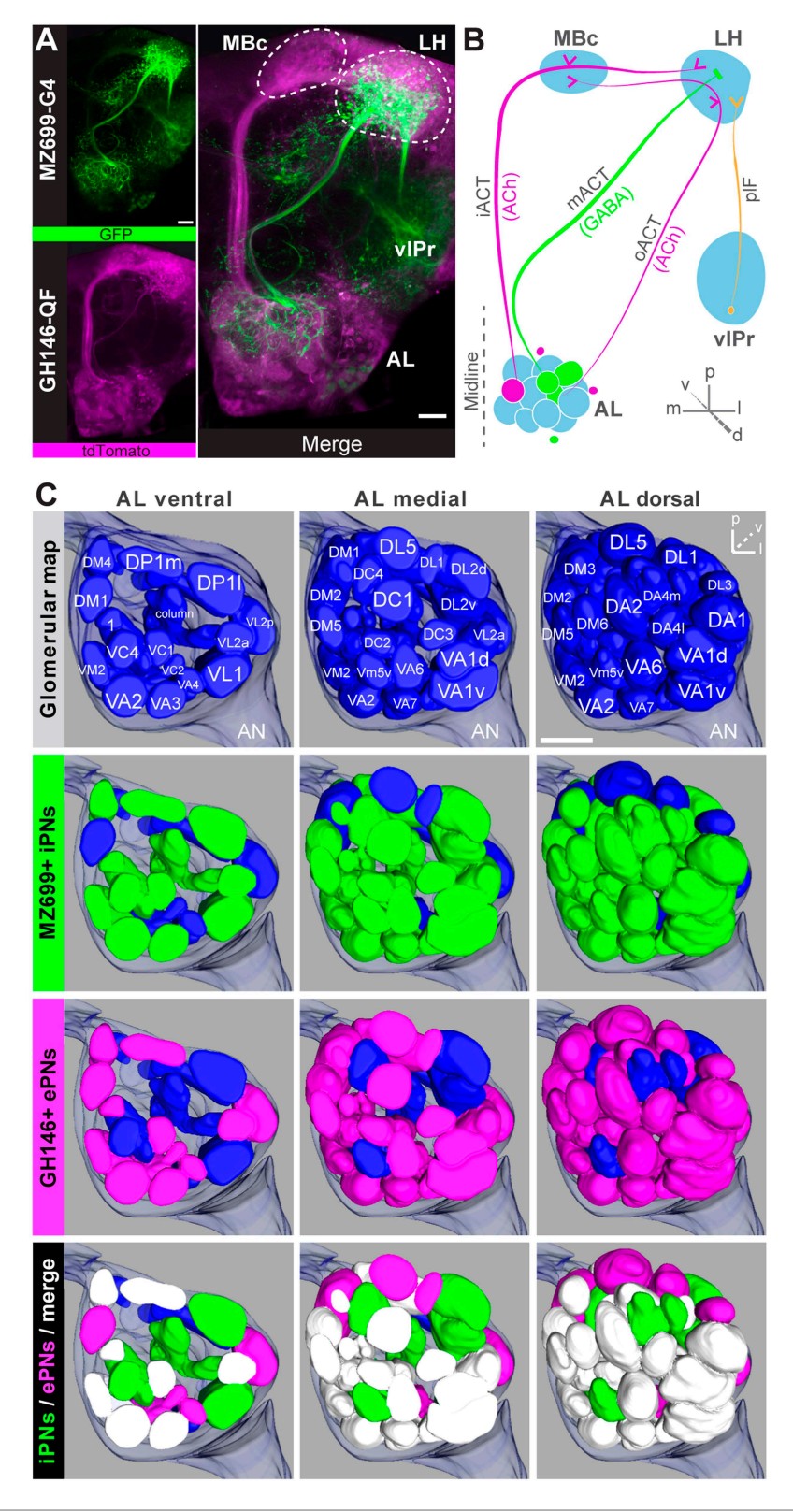

**Figure 1**. Detailed glomerular innervations of excitatory and inhibitory projection neurons in the AL. (**A**) Simultaneous labeling of inhibitory projections neurons (iPNs, labeled by *MZ699-GAL4;G-CaMP*) and excitatory projection neurons (ePNs, labeled by *GH146-QF;tdTomato*) in vivo reveals distinct projections to the lateral horn (LH). All iPNs

*Figure 1. Continued on next page*

*Figure 1. Continued*

bypass the mushroom body calyx (MBc) and innervate the LH exclusively. The MZ699 line labels a few ventrolateral protocerebral neurons (vlPr neurons) projecting via the posterior lateral fascicle (plF) from the ventrolateral protocerebrum (vlPr) to the LH. (**B**) Schematic of the PN connectivity relay from the antennal lobe (AL) to higher brain centers (ePNs in magenta, iPNs in green, and vlPr neurons in orange). (**C**) Above, complete glomerular assignment of the AL neuropil (right AL), labeled with elav-*n*-synaptobrevin:DsRed (END1-2). Below, glomerular innervations of both PN populations related to in vivo images in *Figure 1—figure supplement 2*. Depicted are the ventral level (~−40 μm), the medial level (~−20 μm) and the dorsal view onto the AL. Color annotation: blue glomeruli are not innervated by any of the used GAL4-lines; green glomeruli are innervated by MZ699+ iPNs and magenta by GH146+ ePNs; white glomeruli are innervated by both enhancer trap lines. Scale bar, 20 μm.

The following figure supplements are available for figure 1:

**Figure supplement 1**. Characterization of excitatory and inhibitory projection neurons.

**Figure supplement 2**. Glomerular innervations of ePNs and iPNs.

additional region was recruited. Benzaldehyde elicited no response at very low concentrations, but induced clear activity at median and high concentrations in a third region, which was completely separate from the regions activated by the other two odors. Observed patterns were highly reproducible within one animal and stereotypic among different individuals, as shown for the stimulation with 1-octen-3-ol (*Figure 2D*) as well as other odors (*Figure 2—figure supplement 1*).

Due to the lack of morphological landmarks in the LH, functional data were analyzed using the pattern recognition algorithm Non-Negative Matrix Factorization (NNMF) (*Lee and Seung, 1999*), which automatically extracts spatial areas possessing a common distinct time-course, further termed LH odor response domains (ORDs). The NNMF analysis extracted three clearly reproducible and spatially robust ORDs (*Figure 2E*, see NNMF part in the 'Materials and methods' section). Notably, ORDs occupying common temporal kinetics exhibited highly stereotypic spatial patterns. We termed the ORDs LH-PM (LH-posterior-medial), LH-AM (LH-anterior-medial) and LH-AL (LH-anterior-lateral) according to their anatomical positions. To validate our observations, we extended our stimulus array to 11 additional odorants and applied each at three concentrations. Odorants were chosen according to chemical classes, hedonic valence and biological value. Hence, the odor set included acids, lactones, terpenes, aromatics, alcohols, esters, ketones and the natural blend, balsamic vinegar. Remarkably, analysis of the additional odorants revealed neuronal activity exclusively within the three described ORDs (*Figure 2F*, *Figure 2—figure supplements 2,3*). Furthermore, median NNMF-extracted Ca²⁺ response traces with indicated statistical quartiles illustrate very low variability and highly reproducible LH signals. The LH-PM area chiefly revealed robust odor-evoked responses across concentrations, while the LH-AM and LH-AL were mainly activated at very high odor concentrations by distinct odorants. The global responsiveness within separate ORDs in the LH substantiates our finding of a relatively broad AL input to MZ699+ iPNs which converges into three spatially regionalized and stereotypic LH activity domains.

## iPNs can be divided into two morphological classes

We next investigated if the spatially regionalized odor-evoked response patterns are reflected in the axonal terminal fields of MZ699+ iPNs in the LH. To analyze these neurons at the single neuron level, we performed neural tracing by employing a genetically encoded photoactivatable GFP (PA-GFP) (*Patterson and Lippincott-Schwartz, 2002*; *Datta et al., 2008*; *Ruta et al., 2010*). The photoconversion of all MZ699+ neurons leaving the AL confirmed the homogeneous distribution of iPN neurites in the LH and the sparse innervation of the posterior-lateral region as mentioned above (*Figure 3A*). Next we illuminated PA-GFP in single somata to selectively label individual MZ699+ iPNs from the soma up to the farthest axonal terminals in the LH. Individual iPNs were reconstructed and transformed into a reference brain using the END1-2 background (*Grabe et al., 2014*) to align neurons of different individuals. Based on their innervation pattern in the LH, MZ699+ iPNs could be assigned to two major morphological classes (*Figure 3B,C*). As expected from the extracted ORDs, one iPN group diverged to the LH-PM region (8/25 of iPNs), while a second group extended their axonal terminations within the LH-AM area (10/25 of iPNs). In order to statistically substantiate our observation, we

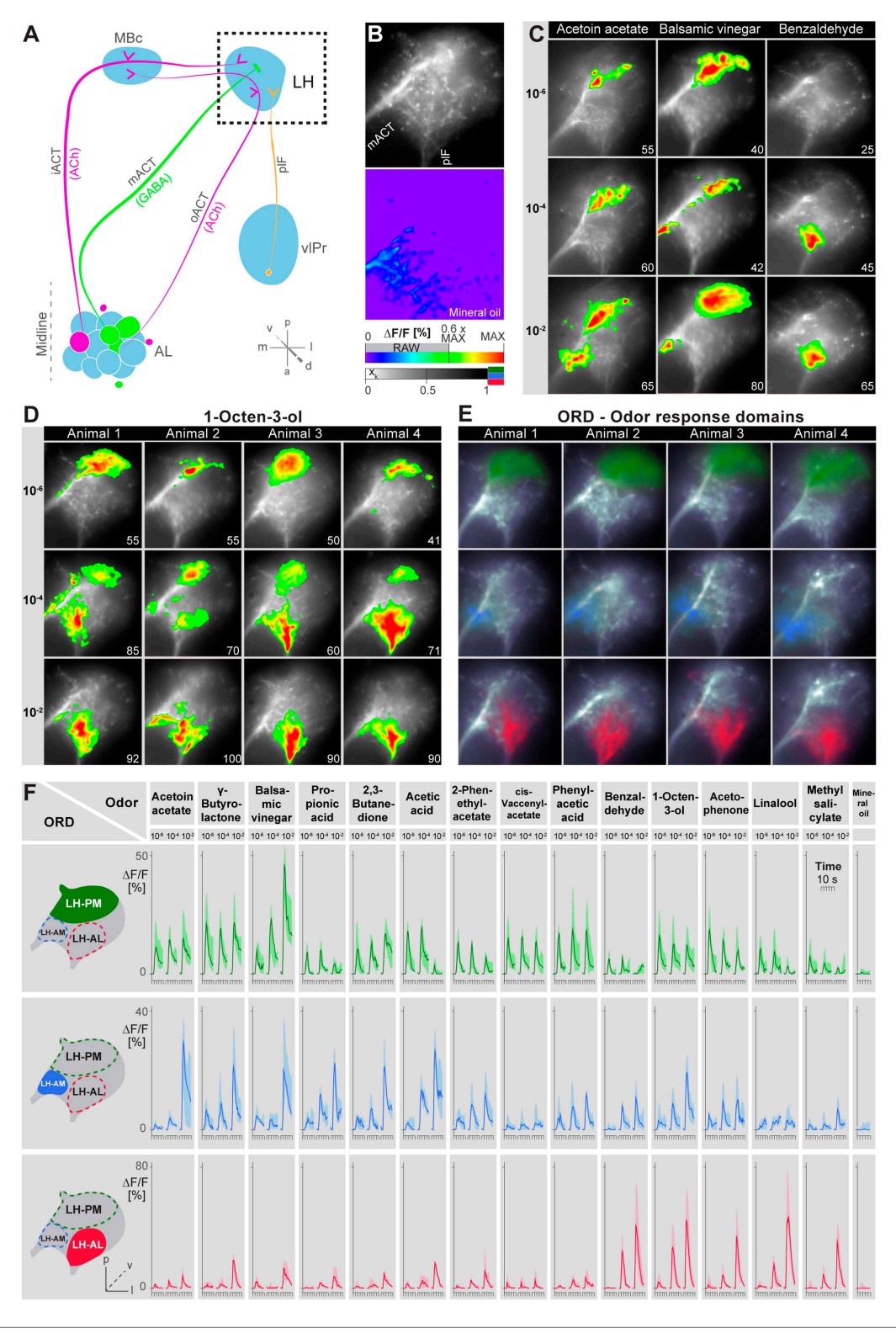

**Figure 2**. Odors evoke specific and stereotypic calcium responses in the LH subdivided into three distinct odor response domains. (**A**) Schematic of the olfactory circuit with the investigated area highlighted. (**B**) RAW image of the LH (top picture) depicting the recorded area of figures (**C**–**E**) and the false color image (bottom picture) during the solvent application. The ΔF/F scale bar applies for all false color-coded pictures; the alpha-bar for the pixel

*Figure 2. Continued on next page*

*Figure 2. Continued*

participation $x_k$ of the indicated colors applies for (**E**–**F**). (**C**) Representative LH Ca²⁺ responses (ΔF/F%) of acetoin acetate, balsamic vinegar and benzaldehyde at three concentrations. Numbers in the lower right corner indicate individual maxima. (**D**) Odor-evoked Ca²⁺ responses (ΔF/F%) are exemplarily depicted for 1-octen-3ol- at three concentrations in four animals. (**E**) NNMF-extracted LH odor response domains (ORD) of four representative animals: three LH ORDs were fully reproducible after being extracted from all measured animals. Domains classified as identical are similarly color-coded: the green ORD is located in the posterior-medial region of the LH (LH-PM); blue, in the anterior-medial (LH-AM), and red in the anterior-lateral LH area (LH-AL). The alpha-bar for green, blue and red shades is placed in (**B**). (**F**) Left, schematic outlines of the LH with indicated ORDs. Right, median activity traces of all odors at three concentrations are depicted for each colored ORD. Shadows represent lower and upper quartiles (n = 6–7).

The following figure supplements are available for figure 2:

**Figure supplement 1**. Odor-evoked activity patterns in the LH are reproducible and stereotypic.

**Figure supplement 2**. Odor-evoked activity patterns in the LH can be reconstructed with five components.

**Figure supplement 3**. Odor-evoked activity patterns in the LH cluster into three components.

performed a cluster analysis based on a similarity score (*Kohl et al., 2013*) of the target areas of all terminals of each iPN in the LH (for details see 'Materials and methods'). The dendrogram of morphological similarity between each individual iPN shows that all, except one iPN, could be clustered according to their target region in either the LH-AM or LH-PM area (*Figure 3D*) which confirms the classification into two major categories. Additionally, we performed a principal component analysis based on the distances of the similarity scores showing that both neuronal classes possess significantly different target areas in the LH (*Figure 3—figure supplement 1A*; p < 0.001, one-way ANOSIM).

We did not observe any clear panglomerular innervations of individual MZ699+ iPNs that spanned the entire AL, consistent with *Liang et al. (2013)*. Instead, MZ699+ iPNs develop mainly oligoglomerular patterns innervating on average 5.4 ± 3.9 glomeruli (mean ± SD), which are not necessarily in close proximity. It is important to note here, that the glomerular innervations of iPNs are rather sparse in comparison to the innervation of ePNs which complicates the identification of truly innervated glomeruli. After classifying all registered neurons according to their LH zones along with their glomerular innervations, we noted a spatial subdivision of MZ699+ iPN dendritic fields in the AL (*Figure 3—figure supplement 2*). Whereas LH-PM iPNs extended dendrites mainly into glomeruli from the ventro- or dorsomedial area of the AL (e.g., DM4, DM2, VM7, VM5d), iPNs targeting the LH-AM zone innervated glomeruli ranging from the ventro- and dorsoanterior to the dorsocentral region (e.g., DC3, VC1, VA6, VL1). We observed that a glomerulus is typically innervated by only LH-PM iPNs or LH-AM iPNs. However, we also found a few cases where a glomerulus can be innervated by both iPN types (e.g., glomeruli D and DC2). In order to analyze whether the two categories of iPNs can also be statistically separated according to their glomerular innervations in the AL, we performed a cluster analysis based on the glomeruli innervated by each individual iPN (*Figure 3E*). Notably, the two iPN classes could be clearly clustered into two groups due to their specific AL innervations. This finding is further supported by a principal component analysis showing that iPNs targeting the LH-PM region innervate a significant different glomerular subset than iPNs that send their axonal terminals to the LH-AM area (*Figure 3—figure supplement 1B*; p < 0.001, one-way ANOSIM). In accordance with our finding of two major iPN categories is the study by *Lai et al. (2008)* who observed several different stereotyped projection patterns of multiglomerular MZ699+ single-cell clones that could be broadly categorized into two groups based on the dendritic and axonal projection patterns. While we observed corresponding innervated areas in the AL, the described target areas in the LH seem to differ. However, due to the lack of 3D reconstruction of the single-cell clone data, the innervation patterns cannot be compared in detail.

In addition to the oligoglomerular iPNs, we observed a few uniglomerular MZ699+ iPNs innervating either glomerulus DA1 or VL1 (4/25 of iPNs), consistent with *Lai et al. (2008)*, which target the LH-AM region (*Figure 3—figure supplement 2*). Moreover, we identified three other MZ699+ neurons that did not innervate the AL and sent their axons through the mACT to the LH and/or the MBc.

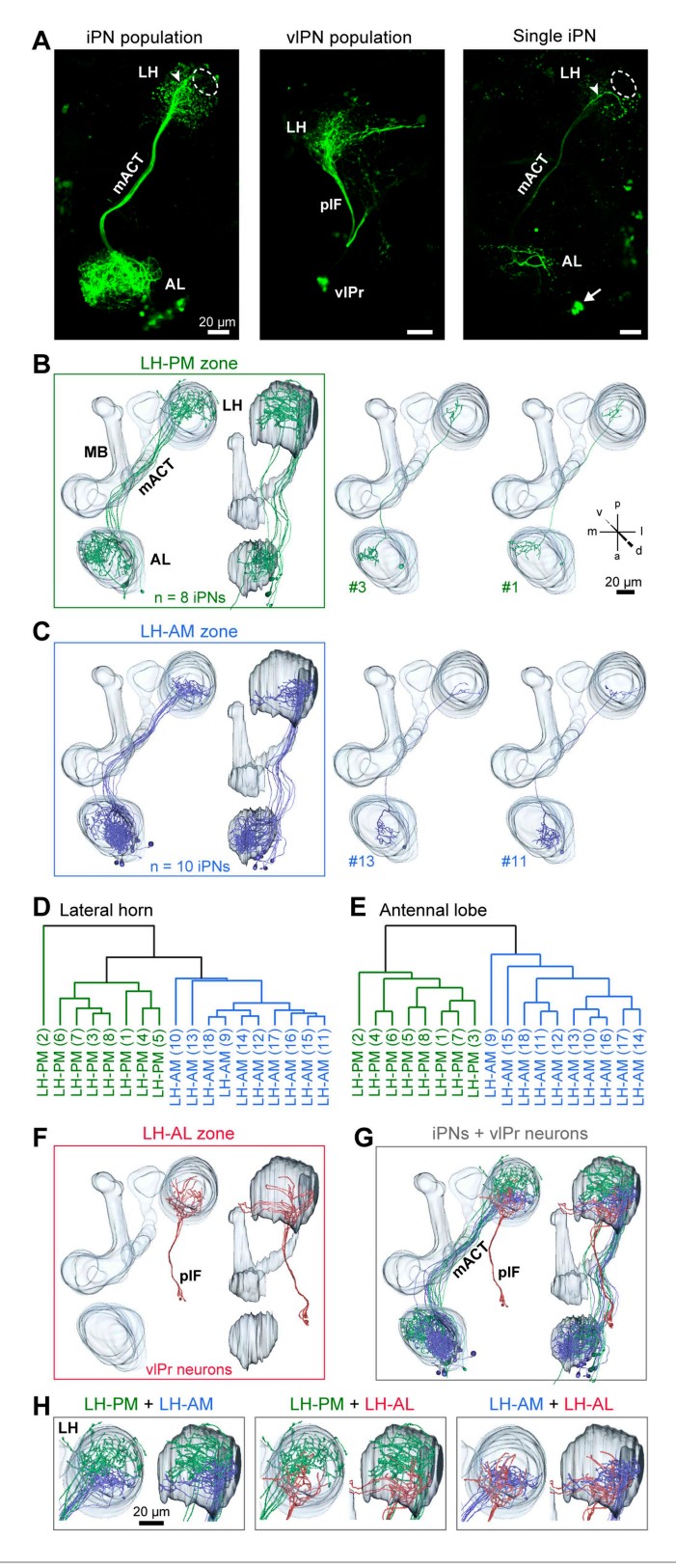

**Figure 3**. iPNs can be classified according to their projection pattern in three distinct LH zones. (**A**) Complete population of MZ699+ iPNs labeled using PA-GFP (left image), the posterior-lateral LH region is encircled, arrowhead indicates the final common projection point of iPN axons. Middle image: photoactivation of all vlPr

*Figure 3. Continued on next page*

*Figure 3. Continued*

neurons of the MZ699-GAL4 line that project from the LH to the vlPr via the plF. Right image: exemplary single iPN, labeled by photoconverting PA-GFP in a single soma (arrow). Scale bar, 20 µm. (**B**) Framed images: neuronal reconstructions of all iPNs projecting to the LH-PM zone (n = 8) with outlined olfactory neuropils. View from dorsal (left) and lateral (right). Right part represents two exemplary registered individual iPNs. (**C**) Neuronal reconstructions of all iPNs projecting into the LH-AM zone (n = 10), images are arranged as in (**B**). (**D** and **E**) Cluster analyses based on the target areas of all terminals of each iPN in the LH (**D**) or based on the innervated glomeruli in the AL (**E**). The dendrograms are split into colored subclusters. Below each dendrogram, each individual iPN is specified according to the labels in *Figure 3—figure supplement 2*. Note, that iPNs can be morphologically clustered according to their target or input regions. (**F**) Neuronal reconstruction of vlPr neurons projecting through the plF to the LH-AL zone. (**G**) Combination of all registered neurons. (**H**) Dual combinations of all registered neurons with their projections in the LH.

The following figure supplements are available for figure 3:

**Figure supplement 1**. iPNs can be morphologically segregated according to their target and input region.

**Figure supplement 2**. Glomerular innervations of individual iPNs.

Since the *MZ699-GAL4* line labels also neurons connecting the LH and the ventrolateral protocerebrum (vlPr) (*Ito et al., 1997*; *Liang et al., 2013*; *Parnas et al., 2013*), we illuminated a small fraction of the posterior lateral fascicle (plF) to target these putative third-order neurons (*Figure 3A*). The plF comprised axons of ventrolateral protocerebral neurons (vlPr neurons), which bifurcated within the LH-AL (*Figure 3F*). Combinations of all registered neuron types within the assigned zones revealed that iPNs of the LH-AM area and vlPr neurons of the LH-AL region intermingle (*Figure 3G,H*).

## Odor response domains contain the activity of distinct neuronal populations

To illustrate higher-order connectivity, we labeled the three major neuron types, that is, MZ699+ iPNs, GH146+ ePNs and vlPr neurons, targeting the LH within the olfactory circuitry using PA-GFP (*Figure 4A*). Since our observed Ca$^{2+}$ responses in the LH-AL region might reflect activity from vlPr neurons rather than iPNs, we dissected the neuronal contributions within each extracted ORD by conducting transection experiments using two-photon laser-mediated microdissection (*Figure 4B*). By transecting the mACT, we aimed at abolishing LH-responses deriving from MZ699+ iPNs, while cutting the plF connection should eliminate potential odor-evoked vlPr neuron activity. To achieve unambiguous and comparable results, functional imaging was performed in both brain hemispheres simultaneously. Immediately after the intact brain areas were imaged, the tracts were selectively transected on one brain side each (*Figure 4C*) and the imaging procedure was repeated. We applied a reduced odor set that elicited activity in all ORDs and performed NNMF for pre- and post-lesion recordings. Transecting the mACT significantly reduced responses in the LH-PM and LH-AM region, whereas LH-AL responses were significantly abolished by plF-ablation (*Figure 4D*). Notably, we observed that LH-AL responses to some odors were significantly increased after mACT transection as a consequence of the suppression of iPN inhibition of vlPr neurons confirming the study by *Liang et al. (2013)*. Hence, activity in the LH-PM and LH-AM domain can be assigned to MZ699+ iPNs, while LH-AL activity is mainly evoked by vlPr neurons (*Figure 4E*).

## iPN activity in the lateral horn mediates flies' attraction to odors

We next addressed the behavioral relevance of MZ699+ iPN activity in the LH for innate odor-guided behavior. To precisely target iPN function, we expressed an RNAi construct against glutamic acid decarboxylase 1 (GADi) to selectively knock-down the GABA synthesis in MZ699+ iPNs (*Figure 5A*). We confirmed the reduction in GABA production via immunostaining (*Figure 5B*). Since vlPr neurons are not GABAergic, they were not affected by the RNAi expression (*Liang et al., 2013*; *Parnas et al., 2013*). Using wild-type flies and parental controls, we conducted T-maze assays (*Tully and Quinn, 1985*; *Chakraborty et al., 2009*) with nine of the odorants applied in functional imaging experiments at medium and high concentrations. Notably, flies with silenced MZ699+ iPN GABA production revealed a neutral or aversive behavioral response to attractive odors, while repellent odors

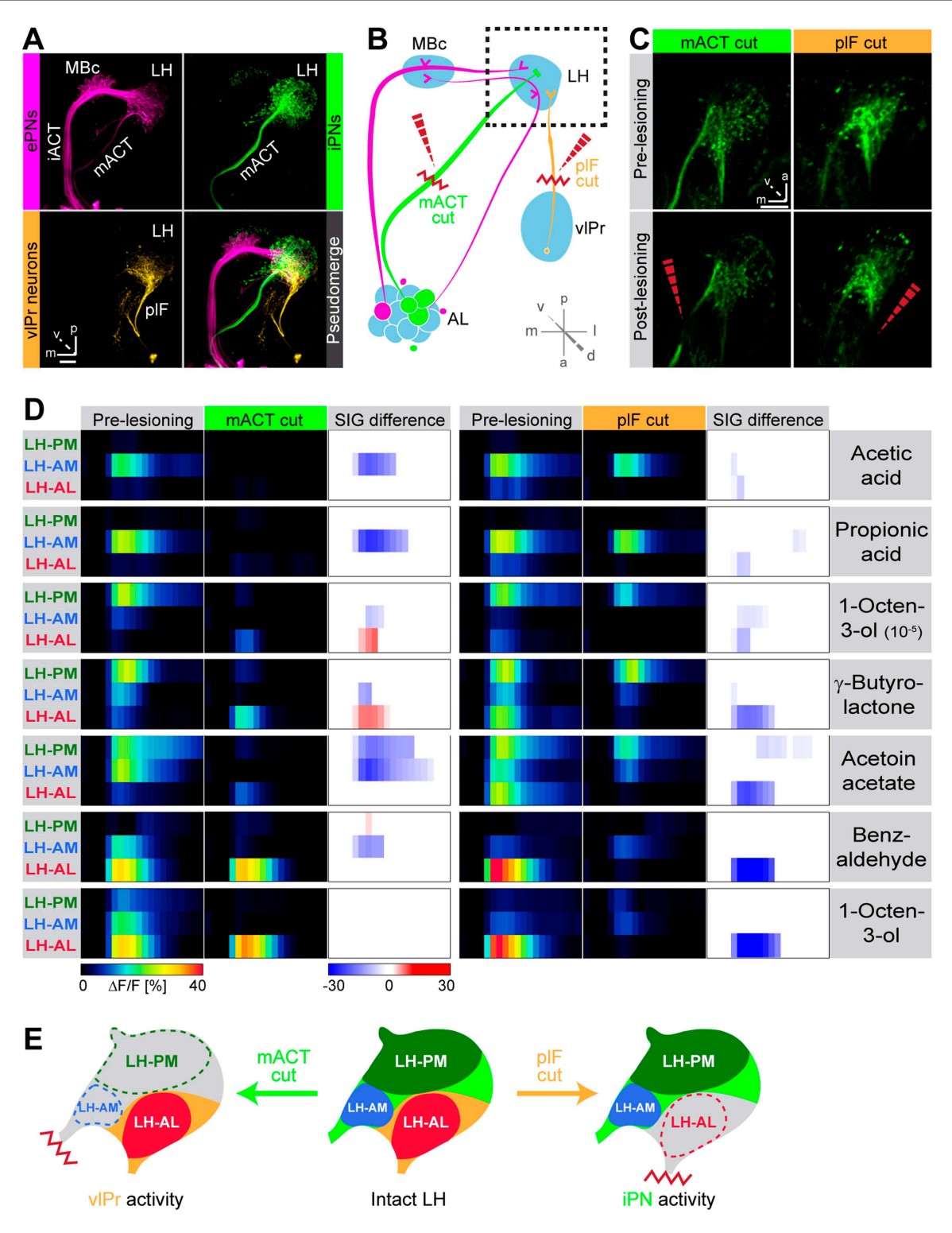

**Figure 4**. Distinct odor response domains in the LH constitute neuronal activity of iPNs and vlPr neurons. (**A**) Representation of all ePNs (magenta) and iPNs (green) labeled by *GH146-GAL4* and *MZ699-GAL4* using PA-GFP, respectively. Photoactivation of vlPr neurons (orange, MZ699-GAL4) connecting the LH and the vlPr via the plF. The overlay image depicts a pseudo-merge image of the different GAL4-driver lines. (**B**) Schematic of the olfactory circuit with integrated layout of the transection experiment. After simultaneous Ca²⁺ imaging of bilateral LHs, the ipsilateral plF and contralateral mACT was transected (red zigzag line) with an infrared laser (dashed red arrow). (**C**) Projection images of a 7 µm stack of the LH area prior and post transection. Left

*Figure 4. Continued on next page*

*Figure 4. Continued*

images, mACT transected; right image, plF transected. The ablated region is indicated by the dashed red arrow. Scale bar, 20 µm. (**D**) Median time traces displaying percental change of ΔF/F values for indicated ORDs prior to post transection of the mACT (green, left) and the plF (orange, right) for different odorants. Significant changes of odor-evoked $Ca^{2+}$ signals due to transection are shown in the column SIG difference. Differences were tested with a two-tailed paired Student's *t* test ($p < 0.05$). Color codes are indicated by the corresponding scale bar below, n = 4–5. Transecting the mACT eliminates $Ca^{2+}$ signals in the LH-PM and LH-AM domain, while lesioning the plF significantly abolishes LH-AL responses. Notably, the LH-AL domain is significantly stronger activated after mACT transection following application of 1-octen-3-ol and γ-butyrolactone. (**E**) Summarized cartoon of the neuron populations contributing to ORD activity prior and post transection of axons of iPNs or vlPr neurons.

evoked an even stronger aversion (*Figure 5C*). To compare the T-maze data more accurately, we calculated the average change of behavioral response indices (RIs) between GADi flies and parental controls (*Figure 5D*). Indeed, all responses changed in a negative direction, indicating MZ699+ iPNs play a crucial role in mediating attraction behavior. The sole exception involved high concentrations of the most repulsive odor, acetophenone, since this odor had already induced maximum aversion. Overall, these experiments reveal a crucial function of MZ699+ iPNs in mediating attraction behavior by releasing GABA in the LH.

## The lateral horn integrates hedonic valence and odor intensity into separate domains

The behavioral effect of the iPN knock-down suggests that MZ699+ iPNs encode positive hedonic valences. To correlate the complete ORD pattern array with innate behavioral preferences, we assigned behavioral RIs for all odors at median and high odor concentrations using the T-maze assay as in our previous experiment (*Figure 6A*). Since extremely low concentrations rarely evoked any behavioral response, we excluded the $10^{-6}$ concentration in this analysis. It is important to note here, that different behavioral assays for testing olfactory preferences in flies might lead to contradictory results. However, the majority of odors used here was also tested in two other behavioral paradigms, the trap assay (*Stökl et al., 2010*; *Knaden et al., 2012*) and the FlyWalk (*Steck et al., 2012*) (pers. comm. M Knaden) and yielded similar results (see *Figure 6—figure supplement 1*). When we plotted median odor-evoked activity in a three-dimensional space defined by the three ORDs, we saw a clear clustering of responses evoked by aversive and attractive odorants (*Figure 6B*). The LH-AL domain, constituted mainly by vlPr neurons, is coding aversive odors, while attractive odors activated only the LH-PM and LH-AM domains that derive from MZ699+ iPNs. This result is in accordance with our finding that iPNs mediate odor attraction.

We next correlated ORD activity to odor valence separately for all ORDs. This evaluation enabled us to analyze iPN and vlPr neuron coding properties apart from each other (*Figure 6C*). As expected, the analysis revealed a significant correlation between positive valence and the LH-PM domain, whereas $Ca^{2+}$ responses in the LH-AL were strongly negatively correlated to hedonic valence. The LH-AM domain exhibited a positive but not significant correlation for odor valence. Remarkably, activity within the LH-PM was totally independent of concentration, whereas activity in both anterior domains was significantly correlated to odor intensity (*Figure 6D*). Hence, MZ699+ iPNs integrate odor attraction information into the LH-PM domain independent of odor intensity, confirming behavioral experiments. Intensity coding is in turn conducted separately by distinct iPNs within the LH-AM domain. In contrast, putative third-order vlPr neurons projecting into the LH-AL area code both negative valence and odor intensity.

Finally, we wondered if this valence-specific LH representation is already reflected at the primary level of olfactory processing. The odor-evoked responses in iPNs are generally similar to those in OSNs (*Wang et al., 2014*), indicating a straight forward transduction of cholinergic OSN responses. We therefore performed functional imaging of odor-evoked $Ca^{2+}$ dynamics at the AL input level by expressing G-CaMP3.0 in OSNs using Orco-GAL4 (*Larsson et al., 2004*) (*Figure 6—figure supplement 2*). In order to compare the activity patterns at both processing levels, we calculated correlation distances for all pair-wise combinations of odor-evoked response patterns and plotted these with respect to maximal ORD pattern similarity in the LH (*Figure 6E*). As expected, odor representations in the LH clearly clustered within three separated parts of the matrix, reflecting our observed ORDs. However, this coding similarity could not predict AL activity patterns, even if the correlation matrix was sorted with respect to pattern similarity in the AL (*Figure 6—figure supplement 3*).

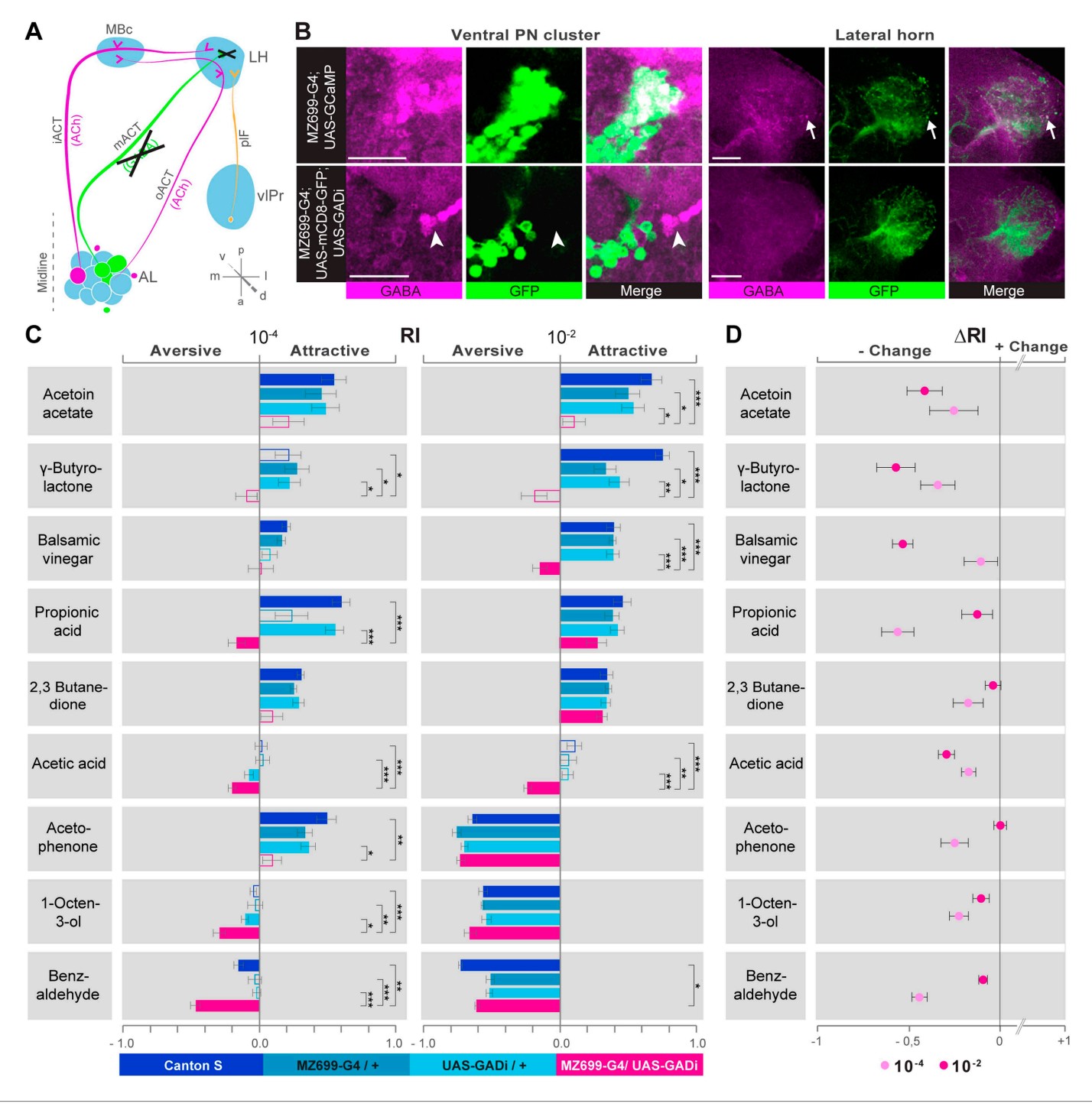

**Figure 5**. iPN GABA release in the LH mediates odor attraction behavior. (**A**) Experimental layout: iPN GABA production was selectively silenced via GADi expression in MZ699+ iPNs; ePN and vlPr neuron activity remained unaffected. (**B**) Immunostaining against GABA and GFP within AL somata (left) and LH neurites (right) of iPNs with intact (top) and silenced GABA production (bottom). GADi flies show GABA signals in somata of iPNs labeled by *GH146-GAL4* only (arrowhead). The arrow head points to an exemplary GABA-positive bouton in the LH. Scale bar, 20 μm. (**C**) Averaged behavioral response indices (RIs) determined with a T-maze assay for wild-type flies (dark blue), parental controls (light blue) and experimental animals (magenta) for nine odorants at two concentrations. Empty boxes display no response (Wilcoxon signed-rank test). Dunn's Multiple Comparison Test was used for global differences in the dataset followed by a posthoc test for selected pairs (p* < 0.05; **p < 0.01; ***p < 0.001). Error bars represent SEM. (**D**) RI differences between GADi flies and averaged parental controls. RI differences are negative for all but one odor indicating that GADi expression shifts odor-guided behavior towards aversion. Error bars indicate SEM.

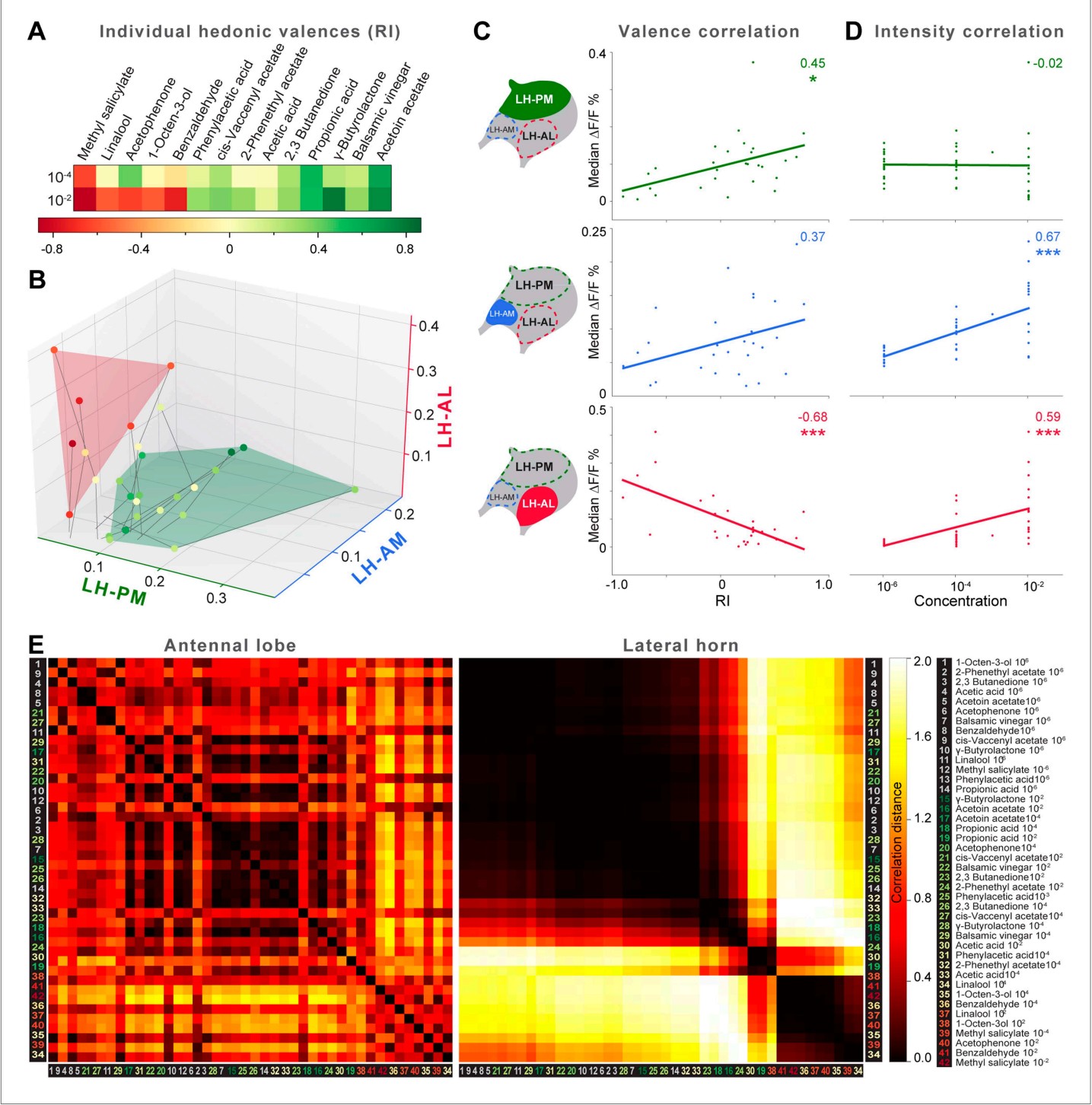

**Figure 6**. Integration of hedonic valence and odor concentration into ORDs. (**A**) Response indices of wild type flies for all odors at median and high concentrations. Odors are sorted from highly aversive (−1, red) to highly attractive (+1, green). (**B**) 3D-scatter plot of median Ca²⁺ responses of all odors based on the three ORDs. Odor-dots are labeled due to their RI shown in (**A**). Same odors at different concentrations are connected with a line: the dot at the end depicts 10⁻², the centered dot 10⁻⁴, and the end of the line 10⁻⁶. Attractive and aversive odor representations form separate clusters. (**C** and **D**) Left, schematic LH outlines with colored ORDs corresponding to data on the right. Correlation score r (upper right corner) between median activity and measured RI in T-maze experiments or odor concentration, respectively, with significance denoted below. Student's t test, *p < 0.05, ***p < 0.001. (**E**) Complete correlation matrices for Ca²⁺ response patterns of OSNs in the AL (left) and iPNs in the LH (right). The odors are arranged according to single linkage clustering of the LH activity patterns. Heatmap color-code refers to the correlation distance scale bar on the right. Correlation distance is

*Figure 6. Continued on next page*

*Figure 6. Continued*

defined as 1 − r, where r is the Pearson correlation coefficient between the response patterns of two odorants. Odor letters are color-coded according to hedonic valence; $10^{-6}$ RI values are labeled in grey (complete list right hand).

The following figure supplements are available for figure 6:

**Figure supplement 1**. Odor valences determined with three different behavioral assays.

**Figure supplement 2**. Calcium responses of OSNs.

**Figure supplement 3**. Correlation matrices for odor-evoked responses in the AL and LH.

## Discussion

We augment our present understanding of the *Drosophila* olfactory circuitry by elucidating the coding properties of a parallel and behaviorally relevant higher-order processing pathway to the LH. Morphological, functional and behavioral approaches provide strong evidence for a functional subdivision of iPNs into neurons coding either odor attraction or odor intensity. Our behavioral experiments reveal that inhibitory properties of iPNs are necessary for innate odor-guided attraction. In addition, we characterize a third neural pathway coding odor repellence.

Do MZ699+ iPNs fulfill anatomical requirements to constitute a distinct processing channel in addition to ePNs? A remarkable anatomical feature of MZ699+ iPNs is their glomerular innervation pattern in the AL. Whereas GH146+ ePNs are uniglomerular and retain the topographic code in their axonal arrangement (*Marin et al., 2002*; *Wong et al., 2002*; *Jefferis et al., 2007*), most MZ699+ iPNs possess oligoglomerular innervations suggesting that these neurons might not convey precise odor-identity information. In addition, MZ699+ iPNs in the AL diverge only into specific glomerular subsets, and so might be pre-determined to selectively extract common features of distinct odors. We have previously shown that the AL map at the PN level exhibits a spatial segregation of valence representation (*Knaden et al., 2012*). Certain glomeruli, which have been classified as aversion coding at the GH146+ ePN level, are omitted by MZ699+ iPNs, whereas most glomeruli classified as attraction coding are particularly densely innervated. These results suggest that within the MZ699+ iPN population, mainly positive odor traits are extracted, whereas the odor information of negative valence is neglected. This conclusion is consistent with the recent finding that one type of LH neurons is receiving input from PNs that mainly innervate glomeruli coding fruity-smelling acetates (*Fisek and Wilson, 2014*) which represent attractive odor cues (*Knaden et al., 2012*). We furthermore demonstrate that the MZ699+ iPN population is split into two major morphological classes possessing a clear spatial segregation in the AL which is strictly maintained within the LH. It has to be kept in mind that we do not cover all iPNs by using *MZ699-GAL4*. Further experiments characterizing the ~6 missing MZ699– iPNs, which are labeled by *GH146-GAL4* (*Wilson and Laurent, 2005*; *Lai et al., 2008*), will elucidate if our assumptions apply for the whole iPN population.

So far only a handful neuroanatomical studies targeting GH146+ ePNs have dealt with the question of how olfactory information is integrated and read out by higher brain structures, in particular the LH (*Marin et al., 2002*; *Wong et al., 2002*; *Tanaka et al., 2004*; *Jefferis et al., 2007*). A recent study that traced the projection pattern of PNs coding ammonia and amines as attractive stimuli and carbon dioxide and acids as repulsive signals suggests that sensory stimuli of opposing valence are represented in spatially distinct areas within the LH (*Min et al., 2013*). In addition the study by *Liang et al. (2013)* showed that MZ699+ iPNs selectively suppress the activity of vlPr neurons to food odors, while pheromone responses were not affected verifying the assumption that different odor features are processed separately. However, functional evidence for a feature-based, spatially segregated activity map in the LH was so far missing.

To unravel the coding properties of MZ699+ iPNs within the LH, we conducted $Ca^{2+}$ imaging experiments of MZ699+ iPNs in the LH to odorants having different hedonic valences and intensities, and could classify the LH into three functional ORDs. Our neuronal tracing and transection experiments validated the LH segmentation into two medial domains that derive from MZ699+ iPNs, and the LH-AL domain formed by vlPr neurons. In line with our observations are morphological studies on ePNs and third-order LH neurons revealing a similarly tight constriction into three zones within the

LH (*Tanaka et al., 2004*), while single-cell labeling combined with image registration resulted in five ePN target zones (*Jefferis et al., 2007*). However, the ePN terminal zones do not necessarily correspond to the target domains of iPNs, since it has recently been shown that MZ699+ iPNs do not inhibit odor responses of GH146+ ePNs (*Liang et al., 2013*) and that the presynaptic sites of iPNs are spatially separated from those of ePNs (*Wang et al., 2014*). Hence both PN populations represent parallel processing pathways that most likely accomplish distinct processing tasks analogous to the honeybee olfactory system which possesses dual olfactory pathways to the higher brain that accomplish parallel processing of similar odors (*Brill et al., 2013*).

Silencing MB function revealed that the LH alone is sufficient for basic olfactory behavior (*de Belle and Heisenberg, 1994*; *Connolly et al., 1996*; *Heimbeck et al., 2001*). Our behavioral results demonstrate that selectively silencing MZ699+ iPNs severely reduced the flies' odor attraction behavior. Hence our results suggest that MZ699+ iPNs are capable of extracting specific features from the combinatorial code emerging in the AL. A behavioral study revealed that silencing MBc neurons impairs odor attraction but not repulsion (*Wang et al., 2003b*). The authors drew the conclusion that the LH is involved in mediating innate repulsion rather than attraction. These results are not necessarily contradictory to ours since some ePNs might activate the LH-AL domain exclusively (i.e., vlPr neurons). On the other hand, *Wang et al. (2003b)* did not include highly concentrated attractive odors. Therefore it is possible that in their experiments, the odor detection threshold was simply reduced, so that only highly concentrated odors, which induced odor aversion, could be distinguished. Our behavioral results, in contrast, revealed the constant influence of the MZ699+ iPNs in mediating attraction for odorants over a range of concentrations.

Our data suggests that odors with opposing hedonic valences are encoded by an interplay of distinct processing pathways. The study by *Liang et al. (2013)* showed that GABA release from MZ699+ iPNs directly inhibits responses of vlPr neurons to food odors as mentioned above. This finding fits well to our observations that iPNs are activated mainly by attractive odors while vlPr neurons are not, likely due to the inhibitory input from iPNs. VlPr neurons are, on the other hand, almost solely activated by repellent odors, which do hardly activate iPNs and therefore do not induce a strong inhibition to vlPr neurons. Repellent odors most likely activate vlPr neurons via ACh release of ePNs which is supported by immunostainings with pre- and postsynaptic markers indicating that vlPr neurons receive input in the LH, while the vlPr represents their major output region (*Parnas et al., 2013*). The vlPr is supposedly also a target of visual neurons from the optic lobe (*Tanaka et al., 2004*) implying that a certain integration of different sensory modalities takes place at this central processing relay. Given that iPNs are inhibitory neurons, the underlying mechanism of odor attraction behavior might therefore be an inhibition of aversive neuronal circuits from the LH to the vlPr that are mainly composed of vlPr neurons. However, this assumption needs to be verified with further experiments elucidating if vlPr neurons are sufficient and necessary to mediate odor aversion.

What is known about odor coding in the LH in other insect species? Notably, in locusts it has been shown that LH neurons receiving convergent PN input appeared to encode stimulus intensity in their net firing rates and in the phases of their spikes (*Gupta and Stopfer, 2012*). Hence these results support the idea that within the LH, general stimulus features such as odor intensity are extracted, which is well in line with our observation of the anterior LH domains whose activity is also significantly correlated to odor intensity. Also in line with our results is a study from honeybees which shows that the representation of different pheromone types is spatially segregated in the LH (*Roussel et al., 2014*), indicating that odors eliciting specific behaviors are coded according to their biological values.

In conclusion, our study provides an important step in unraveling higher olfactory processing mechanisms that are crucial for mediating innate behaviors in *Drosophila*. We provide functional evidence for a feature-based spatial arrangement of the LH decoding opposing hedonic valences and odor intensity. The role of the LH as a center for integrating biological values towards innate decisions by computing conveyed information of two processing pathways is thus expanded.

## Materials and methods

### *Drosophila* stocks

All fly stocks were maintained on conventional cornmeal-agar-molasses medium under L:D 12:12, RH = 70% and 25°C. For wild-type controls *D. melanogaster* of the Canton-S strain was used. Transgenic lines were obtained from Bloomington Stock Center (http://flystocks.bio.indiana.edu/) and Vienna

RNAi stock center (http://www.vdrc.at). Other fly stocks were kindly provided by Kei Ito (*MZ699-GAL4*) and Maria Luisa Vasconcelos (*UAS-C3PA*). The END1-2 fly line is published in *Grabe et al. (2014)*.

## Immunohistochemistry

Whole-mount immunofluorescence staining was carried out as described (*Laissue et al., 1999*; *Vosshall et al., 2000*). Initially brains were dissected in Ringer's solution (130 mM NaCl, 5 mM KCl, 2 mM $MgCl_2$, 2 mM $CaCl_2$, 36 mM saccharose, 5 mM HEPES, [pH 7.3]) (*Estes et al., 1996*) and fixed in 4% PFA in PBS-T (PBS, 0.2–1% Triton-X). After washing with PBS-T brains were blocked with PBS-T, 5% normal goat serum (NGS). Primary antibodies were diluted in blocking solution or PBS-T and incubated at 4°C for 2–3 days. Secondary antibody incubation lasted 1–2 days. Brains were mounted in VectaShield (Vector Labs, Burlingame, CA). The following primary antibodies were used: rabbit α-GABA (1:500) (Sigma), mouse α-GFP (1:500) (Invitrogen). The following secondary antibodies were used: Alexa Fluor 488, goat anti-mouse (1:500); Alexa Fluor 546, goat anti-rabbit (1:500); (all IgG Invitrogen).

## Functional imaging

Fly preparation and functional imaging of the AL was conducted as previously described (*Stökl et al., 2010*; *Strutz et al., 2012*). LH imaging was conducted similarly, except for the higher resolution achieved with a 60× water immersion objective (LUMPlanFl 60×/0.90 W Olympus). The optical plane was ~30 μm below the most dorsal entrance point of the iPN tract into the LH. Binning on the CCD-camera chip resulted in a resolution of 1 pixel = 0.4 × 0.4 μm. For bilateral LH imaging during transection a 20× water immersion objective (NA 0.95, XLUM Plan FI, Olympus) was employed. All recordings lasted 10 s with a frame rate of 4 Hz. Odors included acids (propionic acid, acetic acid), lactones (γ-butyrolactone), terpenes (linalool), aromatics (acetophenone, methyl salicylate, benzaldehyde, phenylacetic acid), alcohols (1-octen-3-ol), esters (acetoin acetate, cis-vaccenyl acetate, 2-phenethyl acetate), ketones (2,3 butanedione) and balsamic vinegar diluted in mineral oil (all from Sigma Aldrich). Odors were applied during frame 8–14 (i.e., after 2 s, lasting for 2 s). Flies were imaged for up to 1 hr, with a minimum inter-stimulus interval of one minute. We selected conventional widefield $Ca^{2+}$ imaging as the method of choice, since we were able to obtain single bouton resolution with this technique.

## Imaging data analysis

Calcium imaging data of AL were analyzed with custom-written IDL software (ITT Visual Information Solutions) provided by Mathias Ditzen as previously described (*Stökl et al., 2010*; *Strutz et al., 2012*). Regarding the $Ca^{2+}$ imaging data in the LH, we repeated recordings of each odor at each concentration two to three times to ensure the reliability of the extracted domain information. To execute NNMF analysis (see below), at least 6–7 valid measurements, that is, animals with repeated identical recordings, were collected for each odor and employed for the analysis. Individual odor measurements were aligned using ImageJ (Fiji) to correct movement artifacts. Fluorescence changes (ΔF/F) for each odor were calculated in relation to background fluorescence using frames 0–6 (i.e., 2–0.5 s before odor application). A Gaussian low-pass filter (ó = 1px) was applied to compensate for remaining movement artifacts and pixel noise. To reduce the computational load, the frame rate was averaged by two consecutive frames, and recordings were spatially down-sampled by a factor of two. The resulting concatenated time-series of the recordings is denoted as measurement matrix Y with element $Y_{t,p}$ being the $t^{th}$ observed value of pixel p.

## NNMF—Non-Negative Matrix Factorization

In contrast to the AL, which consists of highly ordered glomerular subunits, the LH comprises a mainly homogenous neuropil which does not provide spatial or functional landmarks. Therefore, we used the automatic method NNMF to extract $Ca^{2+}$ signals that exhibit common spatial or temporal features. NNMF, like other matrix factorization techniques (e.g., Principal Component Analysis (PCA) and Independent Component Analysis (ICA)), decompose the measurement matrix Y into $k$ components, $Y = \sum_k x_k * a_k^T + R$. The time-course $a_k$ of each component contains a common underlying time-courses of all pixels and each pixel participation $x_k$ declares how strongly each pixel is involved in this time-course. The residual matrix R contains the unexplained data. In order to perform NNMF, we implemented the HALS algorithm in Python including a spatial smoothness constraint ($a_{sm} = 0.1$) (*Cichocki and Phan, 2009*) and an additional spatial decorrelation constraint ($a_{de} = 0.1$) (*Chen and Cichocki, 2005*).

In PCA decomposition is performed such that either timecourses $a_k$ or pixel participation $x_k$ are uncorrelated, whereas ICA aims for timecourses (temporal ICA) or pixel participation (spatial ICA) to be independent. Although spatial ICA is able to segregate signals into functional similar neuropils (*Reidl et al., 2007*), we chose the NNMF approach, because it is known to achieve even a better parts-based representation compared to the more holistic results of PCA or ICA (*Lee and Seung, 1999*). In contrast to PCA and ICA, NNMF constrains both the extracted time-courses and pixel participations to be positive. Positive pixel participation enabled us to make a straightforward physiological interpretation, reading the participation values as the contribution strength of an underlying physiological domain. The restriction to positive time-courses reflects the fact that we did not observe any significant decrease of fluorescence in response to an odor in the original measurement data. For each animal we performed decomposition into $k$ = 5 components. This was sufficient to explain most of the data's variance (88% + 8%, error is standard deviation across individuals). The remaining variance in the residual matrix R contained no additional domains but rather reflected remaining movement artifacts of the measurements (*Figure 2—figure supplement 2*). Of the five components extracted by NNMF, three stood out prominently (*Figure 2—figure supplement 3*): First, they were extracted in all animals at very clearly defined anatomical positions. Second, their responses to stimuli repetitions were highly reproducible in contrast to the other two components, that is, they exhibited a significant (p < $2*10^{-8}$, $t$ test) higher trial-to-trial correlation of 0.72 ± 0.20 in contrast to 0.52 ± 0.26 for the remaining components; third, the odorant spectra of their responses were characteristic across animals.

Though we cannot completely rule out that the remaining components of the factorization are ORDs of their own, there are several indications that they are not. On the one hand, they exhibit a lower trial-to-trial correlation than the three selected components. Second, those components did not consistently appear at similar anatomical position. Third, they were spatially overlapping with the selected three components. Instead of independent ORDs, these regions might convey fluorescence changes independent of odor stimulation or an overlapping region of two of the reliable ORDs. A validation of our NNMF-based results with spatial ICA yielded very similar, but slightly worse results. Whereas the three reliable ORDs from NNMF were also extracted in spatial ICA, the two remaining components exhibited much higher variability than when obtained with NNMF. Hence, we conclude that the LH area comprising MZ699+ neurons consists of three ORDs. We labeled those three components according to the anatomical position of their pixel participation within the LH.

## Statistical analysis of imaging data

To determine the coding properties of extracted odor response domains (ORDs), we calculated the mean response of each animal within a time window of 1–4 s after stimulus onset. Hence, median responses over all animals defined the standard stimulated response $r_{ORD}^o$ of an ORD to an odor $o$. Initially, regions were evaluated individually, and correlations were calculated between standard response spectra and the behavioral response index (RI), or odor concentration, respectively, using the 'linregress' function of the Python scipy.stat module. To analyze the combined ORD representations of odor patterns $p_o = \left( r_{PM}^o, r_{AM}^o, r_{AL}^o \right)$ we calculated for all odor pairs the pattern similarity as correlation distance $d_{o1,o2} = 1 - corr(p_{o1}, p_{o2})$. In order to visualize the correlation matrix in a comprehensible way, we then arranged odors according to the single linkage clustering of the Python scipy.cluster.hierarchy module. To compare the representation in the LH to those of the AL, we applied the same procedure to the dorsal glomerular odor activation pattern.

## 2-Photon photoactivation and neuronal reconstructions

For in vivo photoactivation experiments, 1–6 day old flies (genotype: END1-2,UAS-C3PA;MZ699-GAL4) were dissected as in the imaging experiments except that tracts of the salivary glands were cut to prevent movement. Photoactivation was accomplished via continuous illumination with 760 nm for 15–25 min. After a 5-min break to permit full diffusion of the photoconverted molecules, 925 nm z-stacks of the whole brain were acquired and subsequently used for neuronal 3D-reconstruction. For all 3D reconstructions, the segmentation software AMIRA 4.1.1 & 5.3.3 (FEI Visualization Sciences Group, Burlington, MA) was used. Neurons of different individuals were embedded into the reference brain using a labelfield registration as previously described (*Rybak et al., 2010*). Briefly, segmented labels of brain neuropils (AL, MBc, LH) were registered onto a reference brain image using affine registration followed by elastic warping. In a second step, the calculated

transformation matrix was applied to the respective neuron morphology that was then aligned to the reference brain image.

For morphological analysis of reconstructed iPNs, we first determined all terminal points of each iPN in the LH area. For each combination of terminals we calculated a similarity score (*s*) in analogy to (*Kohl et al., 2013*) as follows:

$$s(t_1, t_2) = \sqrt{e^{-\Delta(t_1,t_2)^2/2\sigma^2}},$$

where *t* is the terminal position, Δ*(t1,t2)* is the Euclidean distance and $\sigma$ is a free parameter that determines how close in space terminal points must be to be considered similar; analogue to *Kohl et al. (2013)* we set this parameter to 3 µm. Finally we calculated the pairwise similarity score between two neurons as their average all-to-all terminal similarity scores, normalized to their self-scores as follows:

$$S(n_1, n_2) = \sum_{t_1, t_2} s(t_1, t_2) / \sqrt{\sum_{t_1} s(t_1, t_1) * \sum_{t_2} s(t_2, t_2)}$$

Effectively this quantifies the relative overlap of the target area of all pairs of iPNs. For clustering, the similarity scores were converted to distances (i.e., *1-S*) and a hierarchical clustering was performed using UPGMA method. Principal component analysis and one-way ANOSIM was performed using the statistical software PAST 3.x (Paleontological statistics software package for education and data analysis).

## 2-Photon-mediated transection

Transections of either the plF tract or the mACT were conducted in one brain hemisphere, each of the same fly. The target area was monitored with 925 nm and chosen to be close to the LH but distant enough not to affect neurites ramifying in the LH neuropil. For both tracts, lesioned areas had an average size of 34 µm and were illuminated with short pulses of 710 nm every 40 ms for 250 ms in 60 (plF) to 80 (mACT) cycles in a single focal plane. After a fast z-stack with 925 nm to confirm complete lesion, a 5-min neuronal recovery interval followed before continuing the imaging procedure. Data were analyzed using NNMF.

## Image acquisition

Photoactivation and transection procedures as well as image acquisition following immunohistochemistry were accomplished with a 2-photon confocal laser scanning microscope (2PCLSM, Zeiss LSM 710 NLO) equipped with a 40× (W Plan-Apochromat 40×/1.0 DIC M27, Zeiss) or 20× (W N-Achroplan 20×/0.5 M27, Zeiss). The 2PCLSM was placed on a smart table UT2 (Newport Corporation, Irvine, CA, USA) and equipped with an infrared Chameleon Ultra diode-pumped laser (Coherent, Santa Clara, CA, USA). Z-stacks were performed with argon 488 nm and helium-neon 543 nm laser or the Chameleon Laser 925 nm (BP500-550 for G-CaMP and LP555 for DsRed/tdTomato) and had a resolution of 1024 or 512 square pixels. The maximum step size for immuno-preparations or single neuron projections was 1 µm and for AL reconstructions 2 µm.

## Behavioral assay

Flies carrying P[GAD1-RNAi];P[MZ699-GAL4] were crossed just before the experiment to prevent dosage compensation effects. T-maze experiments were performed as described (*Stensmyr et al., 2012*). WT, parental controls (P[GAD1-RNAi] or P[MZ699-GAL4]) and test flies carrying both insertions were tested separately under identical conditions. The response index (RI) was calculated as (O-C)/T, where O is the number of flies in the odor arm, C is the number of flies in the control arm, and T is the total number of flies used in the trial. Hence, the RI ranges from −1 (complete avoidance) to 1 (complete attraction). Each experiment was carried out on 30 flies and was repeated 12 times. Dunn's Multiple Comparison Test was used for global differences in the dataset. Whenever the Multiple Comparison Test was significant (i.e., p < 0.05), a posthoc test for selected pairs was performed, that is, between the GADi-flies and the other three control lines as we were not interested in differences among the different control lines. All RI were tested against 0 (no response) by using the Wilcoxon-rank-sum test.

## Acknowledgements

We thank Silke Trautheim, Regina Stieber, Linda Gummlich and Sascha Bucks for excellent technical assistance and Emily Wheeler for editorial assistance.

# Additional information

## Competing interests

BSH: Vice president of the Max Planck Society, one of the three founding funders of *eLife*, and a member of *eLife's* Board of Directors. The other authors declare that no competing interests exist.

## Funding

| Funder | Grant reference number | Author |
|---|---|---|
| Bundesministerium für Bildung und Forschung | | Silke Sachse |
| Max-Planck-Gesellschaft | | Bill S Hansson |
| Deutsche Forschungsgemeinschaft | Priority program SPP1392 (SCHM2474/1-1 and 1-2) | Michael Schmuker, Jan Soelter |

The funders had no role in study design, data collection and interpretation, or the decision to submit the work for publication.

## Author contributions

AS, Conception and design, Acquisition of data, Analysis and interpretation of data, Drafting or revising the article; JS, Analysis and interpretation of data, Drafting or revising the article; AB, VG, JR, Acquisition of data, Analysis and interpretation of data; AF, Performed behavioral experiments; MK, Supervision of Abu Farhan, Analysis and interpretation of data; MS, Supervision of Jan Soelter, Analysis and interpretation of data; BSH, Provided intellectual and financial support, Conception and design; SS, Conception and design, Analysis and interpretation of data, Drafting or revising the article

## Author ORCIDs

Amelie Baschwitz, http://orcid.org/0000-0003-0614-1667
Michael Schmuker, http://orcid.org/0000-0001-6753-4929

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
