## [Decision Letter]

Thank you for sending your work entitled “Decoding Odor Quality and Intensity in the Drosophila brain” for consideration at *eLife*. Your article has been favorably evaluated by a Senior editor, Mani Ramaswami (Reviewing editor), and 2 reviewers, one of whom, Nitin Gupta, has agreed to reveal his identity.

The Reviewing editor and the reviewers discussed their comments before we reached this decision, and the Reviewing editor has assembled the following comments to help you prepare a revised submission.

This study examines the role of a major class of antennal lobe projection neurons, the iPNs, and their targets in the lateral horn. Several recent studies have begun to classify the function of the iPNs and have shown a role in odor discrimination, in inhibiting food odor processing, and in pheromone-mediated behaviors. This study comprehensively examines the anatomy and functional properties of iPNs and makes several important contributions, including that (1) iPNs may be functionally segregated into two different classes based on glomerular inputs, targeting region in the lateral horn and response properties; (2) they are required for olfactory attraction; and (3) they likely inhibit third order neurons that mediate aversion. Overall, this is a very rigorous, nicely executed study that makes important contributions to odor processing in the lateral horn.

Although some of its excitement has been stolen by other recent papers covering iPNs([30]; [40]; [63]; [13], the current study additionally provides the most detailed, glomerulus-level mapping of iPNs in the AL, provides a description of three projection zones of iPNs in the LH, and provide some evidence that these zones correspond to behavior. Another important finding is that blocking GABA-synthesis in iPNs makes the odors more aversive. These results suggest that the inhibition from iPNs usually keeps the aversive pathways under check. This is in agreement with recent work from Liqun Luo's lab (Liang et al), which showed that vlPR neurons form an aversive pathway inhibited by iPNs. The functional imaging and microdissection experiments performed by the authors provide additional (but indirect) evidence to support the same conclusion. Thus, this work reports discoveries and contributions appropriate for publication in *eLife.*

However, there are some issues that need to be addressed prior to publication.

1) In Figure 3, the LH-AL and LH-AM zones are well separated, while in Figure 4, they seem nearly overlapping. Figure 4: It seems that the reconstructions shown in panels B and C were chosen based on their projections into AM or PM zones. The authors have not shown that all iPNs can be unambiguously classified as belonging to these two categories. Is it true that no iPN branches into both zones?

The authors should use their single cell clone data to rigorously test whether lateral horn projections cluster into classes.

2) The authors also comment that AL arborizations are different for the two types of iPNs without providing any analysis–this is important to do. Cluster analysis of morphology parameters coupled with PCA should be straightforward.

3) It is not always clear that some of the work in this manuscript is consistent with published studies or replicates published studies. The authors should be careful to credit previous studies.

For example:“GAD1 (glutamic acid decarboxylase) in situ hybridizations imply GABA synthesis (39), which we verified via immunostaining”. GABA Ab staining is shown in Parnas 2013.

Dendritic and synaptic regions on iPNs have been published several times (okada 09, Liang 2013, Parnas 2013) but the text states: “To determine the polarity of both PN populations, we expressed UAS141Dα7:mcherry to tag postsynaptic input sites by labeling acetylcholine receptors (AChR) (Figure 1), and UAS- Syt:HA to label presynaptic terminals (Figure 1).” MZ699 synaptobrevin and Denmark staining is shown in Parnas 2013, and nSyb or syt staining in Okada 09 and Liang 13, so the polarity of iPNs is known. The polarity of ePNs is also known.

In addition, Lai 2008 did single cell MZ699 clones and noticed regionalization in the lateral horn similar to the single-cell clone studies in this manuscript.

4) Figure 1 does not seem essential and should be justified by the authors. It would better given prior publications to begin with Figure 2, which also contains the summary info from previous studies.

5) “How these crucial functions are accomplished within the olfactory system remains unknown”. The authors should modify this sentence to acknowledge the large body of work in olfaction related to feature detection, both in insects and in vertebrates.

6) In figure supplement 1B, why only a small fraction of the MZ699+ somata in AL appear GABA-positive?

7) Figure 1: There may be weak Syt:HA expression in the AL for mz699+ neurons. Can the authors completely rule of the possibility of synaptic release in the AL? Perhaps a more quantitative analysis would help.

8) Are the activation patterns of iPN LH projections stereotyped among individuals for the two attractive and the one aversive odor tested above? Why is the reproducibility demonstrated using an unrelated odor (1-octen-3-ol) rather than the same three odors?

9) The authors claim that “analysis of the additional odorants revealed neuronal activity exclusively within the three described 'ORDs' (Figure 3)”. Does it mean that there was no activity outside the three ORD regions? If so, no data has been shown to support this claim.

10) The observation that LH-AM and LH-AL were activated only at high odor concentrations could be because these areas had a lower density of branches than LH-PM, and therefore required stronger activation to clear the threshold for detection. Can the authors rule out this possibility?

11) If the three neurons did not innervate the AL, why call them “iPNs”?

12) Benzaldehyde seems to induce the same type of preference (repulsion) for both concentrations in the figure; opposite of what the authors say in the text.

13) In GADi flies, the responses to the two concentrations are different for several odors (vinegar, propionic acid, butadione, acetophenone), so the claim that perception of intensity was impeded is not justified based on the presented data and should be removed from this section. The results are better explained by the hypothesis that GABA blockage makes odors more repulsive (and ceiling effects in behavioral tests).

14) What was the rationale for using Dunn's Selected Pairs test in some cases, when Dunn's Multiple Comparison was used in others?

15) How do the authors' findings on odor coding in lateral horn compare with the findings from other insect species such as honeybees and locusts? Locusts also appear to have odor concentration-dependent responses in the lateral horn.

16) Valence as determined by behavior, seems to be dependent on the exact behavioral assay. Thus for most odorants (except for the rare extreme cases) it could be difficult to assign a clear valence. This would make it makes difficult to establish where absolute valence is encoded in the lateral horn. The authors should consider this issue and discuss potential difficulties in unambiguously defining the “valence” of an odor, as otherwise the literature may be full of contradictory observations.

---

## [Author Response]

*1) In*
Figure 3*, the LH-AL and LH-AM zones are well separated, while in*
Figure 4*, they seem nearly overlapping.*
Figure 4*: It seems that the reconstructions shown in panels B and C were chosen based on their projections into AM or PM zones. The authors have not shown that all iPNs can be unambiguously classified as belonging to these two categories. Is it true that no iPN branches*
*into both zones?*

*The authors should use their single cell clone data to rigorously test whether lateral horn projections cluster into classes*.

We agree with the reviewers and addressed the concern by adding the following data:

A) We photoactivated and reconstructed 4 further iPNs, which are now included into the table in Figure 3—figure supplement 2, resulting in 8 iPNs for the LH-PM area and 10 iPNs for the LH-AM area. Hence we characterized in total 25 MZ699+ PNs.

B) We included the reconstructions of all iPNs into the reference brain in Figure 3 instead of showing only a subgroup of neurons.

C) In order to rigorously test whether all iPNs can be classified into two morphological categories, we performed a cluster analysis based on the target areas of all terminals of each iPN in the lateral horn. To this end we first determined all terminal points of each iPN in the lateral horn area. For each combination of terminals we calculated a similarity score (*s*) in analogy to Kohl et al. (Cell, 2013). Finally we calculated the pairwise similarity score between two neurons as their average all-to-all terminal similarity scores, normalized to their self-scores. Effectively this quantifies the relative overlap of the target area of all pairs of iPNs. For clustering, the similarity scores were converted to distances and a hierarchical clustering was performed using UPGMA method. Using this cluster analysis all iPNs, except one, could be clustered according to their target region in either the LH-AM or LH-PM area in the lateral horn. The cluster analysis is incorporated into Figure 3.

D) We performed a principal component analysis on the distances and a one-way ANOSIM (Figure 3—figure supplement 1) showing that both neuronal classes are statistically different based on their target areas in the lateral horn.

*2) The authors also comment that AL arborizations are different for the two types of iPNs without providing any analysis–this is important to do. Cluster analysis of morphology parameters coupled with PCA should be straightforward*.

We agree with the reviewers and added a cluster analysis as well as a principal component analysis showing that both categories of iPNs innervate different subsets of glomeruli in the antennal lobe and can therefore be morphologically clustered. We also performed a one-way ANOSIM revealing that this difference is highly significant between the two iPN classes. We incorporated the cluster analysis into Figure 3 and the PCA including the ANOSIM into Figure 3—figure supplement 1.

*3) It is not always clear that some of the work in this manuscript is consistent with published studies or replicates published studies. The authors should be careful to credit previous studies*.

*For example:“GAD1 (glutamic acid decarboxylase) in situ hybridizations imply GABA synthesis (*[39]*), which we verified via immunostaining”. GABA Ab staining is shown in Parnas 2013*.

*Dendritic and synaptic regions on iPNs have been published several times (okada 09, Liang 2013, Parnas 2013) but the text states: “To determine the polarity of both PN populations, we expressed UAS141Dα7:mcherry to tag postsynaptic input sites by labeling acetylcholine receptors (AChR) (*Figure 1*), and UAS- Syt:HA to label presynaptic terminals (*Figure 1*).” MZ699 synaptobrevin and Denmark staining is shown in Parnas 2013, and nSyb or syt staining in Okada 09 and Liang 13, so the polarity of iPNs is known. The polarity of ePNs is also known*.

*In addition, Lai 2008 did single cell MZ699 clones and noticed regionalization in the lateral horn similar to the single-cell clone studies in this manuscript*.

We have modified our manuscript following the reviewers’ suggestions in order to ensure that we accurately refer to previously published data. First, we removed our GABA immunostaining from the manuscript, since the *in situ* hybridizations performed by [39] showing that the majority of MZ699+ iPNs are GAD1-positive has already been verified via immunostaining by [40] and [30] as the reviewer mentioned and do not need to be confirmed another time. We refer to both studies in the Results part. Second, we completely removed our data regarding the polarity of iPNs and ePNs and cite the published data instead. Third, we mention and discuss the study by [26] who also observed a categorization of MZ699+ iPNs using single-cell clone data in the corresponding Results section.

*4)*
Figure 1
*does not seem essential and should be justified by the authors. It would better given prior publications to begin with*
Figure 2*, which also contains the summary info from previous studies*.

We agree and therefore begin the paper now with the original Figure 2 as the reviewers suggested.

*5) “How these crucial functions are accomplished within the olfactory system remains unknown”. The authors should modify this sentence to acknowledge the large body of work in olfaction related to feature detection, both in insects and in vertebrates*.

We agree and therefore modified the sentence to address our main question. In order to acknowledge the various studies that have investigated feature extraction in the olfactory system, we cite and discuss them in our Discussion section.

*6) In figure supplement 1B, why only a small fraction of the MZ699+ somata in AL*
*appear GABA-positive?*

We observed that *MZ699-GAL4* is not solely expressed in iPNs and vlPr neurons, but labels also neurons that innervate the subesophageal ganglion in accordance with [30]. Since these SOG neurons are GABA-negative, whose somata are located below the iPN cluster, they appear as unlabeled somata in our GABA staining. However, our aim was to confirm the *in situ* hybridizations performed by [39] showing that on average 80% of mACT iPNs labeled by *MZ699-GAL4* are GAD1-positive. Since [30] and [40] already confirmed that the majority of iPNs is GABAergic, we decided that these data do not need to be affirmed another time and removed our GABA staining.

*7)*
Figure 1*: There may be weak Syt:HA expression in the AL for mz699+ neurons. Can the authors completely rule of the possibility of synaptic release in the AL? Perhaps a more quantitative analysis would help.*

We agree with the reviewers that the image shows weak signals for Syt-HA. Interestingly, also [39] observed a weak *n*-syb-GFP signal of MZ699+ neurons in the AL, which is – as in our case – much weaker than in the LH. The same holds true for Liang et al. (2014) who observed Syt-HA mainly in the LH, while it was largely, not entirely, absent in the AL. To get a more precise staining we also performed vibratome immunostainings that confirm our previous experiment. Both immunos detect an HA tag, that is fused to the presynaptic protein Synaptotagmin. Since the weak signals in the AL appear to be located in the axonal bundle that projects into the mACT, but not in glomerular structures, we presume that the protein or a HA carrying precursor might be transported in the appearing region, but not specifically be targeted to synaptic boutons. However, a weak GABA release of iPNs in the AL cannot be excluded by any of the studies so far. Nevertheless, as written above, we deleted the entire Figure 1 and cited all corresponding papers instead.

*8) Are the activation patterns of iPN LH projections stereotyped among individuals for the two attractive and the one aversive odor tested above? Why is the reproducibility demonstrated using an unrelated odor (1-octen-3-ol) rather than the same*
*three odors?*

The odor 1-octen-3-ol was chosen as a representative example because all three ORDs are activated by this odor over the range of concentration. Moreover 1-octen-3-ol has been employed as a comparative standard odor (due to its comprehensive ORD pattern) in all imaged animals. However, we included now the activation patterns of four animals to the three additional odors acetoin acetate, balsamic vinegar and benzaldehyde as the reviewers requested (Figure 2—figure supplement 1). All three odors were highly reproducible within one animal and stereotypic among different individuals. In addition, the stereotypy for all odors is shown in Figure 2, where the shadow depicts the standard deviation for the odor-evoked responses.

*9) The authors claim that “analysis of the additional odorants revealed neuronal activity exclusively within the three described 'ORDs' (*Figure 3*)”. Does it mean that there was no activity outside the three ORD regions? If so, no data has been shown to support this claim*.

In the initial analysis, we performed decomposition for each animal into five components, since five components were already sufficient to explain 88% ± 8 of the data’s variance. The remaining variance contained no additional domains but rather reflected remaining movement artifacts of the measurements. We added now a supplemental figure showing that the activity patterns of three animals to three odors can be very well reconstructed by using these five components (Figure 2—figure supplement 2) without any remaining stimulus related activity. When we further analyzed these five components, we observed that three of them stood out prominently (as shown now in Figure 2—figure supplement 3). First, they were extracted in all animals at very clearly defined anatomical positions. Second, their responses to stimuli repetitions were highly reproducible in contrast to the other two components, i.e. they exhibited a significant (p<2*10^-8^) higher trial-to-trial correlation. Third, the odorant spectra of their responses were characteristic across animals. Although we cannot completely rule out that the two remaining components of the NNMF are ORDs of their own, there are several indications that they are not. On the one hand, they exhibit a lower trial-to-trial correlation than the three selected components. Second, those components did not consistently appear at similar anatomical position. Third, they were spatially overlapping with the selected three components. Instead of independent ORDs, these regions might convey fluorescence changes independent of odor stimulation or an overlapping region of two of the reliable ORDs. A validation of our NNMF-based results with spatial ICA also extracted the three reliable ORDs, while the two remaining components exhibited even higher variability than with NNMF.

*10) The observation that LH-AM and LH-AL were activated only at high odor concentrations could be because these areas had a lower density of branches than LH-PM, and therefore required stronger activation to clear the threshold for detection. Can the authors rule out*
*this possibility?*

We quantified several morphological parameters of the two neuronal populations in the lateral horn, such as the neuronal volume per µm^3^ as a measure for innervation density, the surface per µm^2^, the number of branches, the lengths of the neuron as well as the total number of terminals using the neuronal reconstructions in AMIRA. However, we could not find significant differences for any of these parameters between our LH-AM and LH-PM neurons indicating that the detection threshold should be equal.

*11) If the three neurons did not innervate the AL, why call*
*them “iPNs”?*

We agree and renamed these neurons to “other MZ699+ neurons”.

*12) Benzaldehyde seems to induce the same type of preference (repulsion) for both concentrations in the figure; opposite of what the authors say in the text*.

We apologize for the confusion here. We intended to say that benzaldehyde evoked a different strength of repulsion for the two concentrations in wildtype flies, while GADi-flies showed an equal strong repulsion independent of the benzaldehyde concentration. However, we decided in general to remove our statement that intensity perception was impeded in GADi flies (see next comment).

*13) In GADi flies, the responses to the two concentrations are different for several odors (vinegar, propionic acid, butadione, acetophenone), so the claim that perception of intensity was impeded is not justified based on the presented data and should be removed from this section. The results are better explained by the hypothesis that GABA blockage makes odors more repulsive (and ceiling effects in behavioral tests)*.

We agree with the reviewers and have therefore removed our conclusion that GADi flies reveal an impaired intensity perception.

*14) What was the rationale for using Dunn's Selected Pairs test in some cases, when Dunn's Multiple Comparison was*
*used in others?*

We used the Dunn’s multiple comparison test for global differences in the dataset. Whenever the multiple comparison test was significant (i.e. p < 0.05), we performed a posthoc test for selected pairs, i.e. between the GADi-flies and the other three control lines as we were not interested in differences among the different control lines. We clarified this statistical method now in more detail to avoid confusions. In addition, we verified all statistical tests once again. While doing that, we realized that an error occurred in our first version, which we have corrected now. We are therefore grateful to the reviewers for questioning the statistics.

*15) How do the authors' findings on odor coding in lateral horn compare with the findings from other insect species such as honeybees and locusts? Locusts also appear to have odor concentration-dependent responses in the lateral horn*.

We added to the Discussion a paragraph summarizing the results from locusts and honeybees regarding odor coding in the lateral horn and compared these findings to ours.

*16) Valence as determined by behavior, seems to be dependent on the exact behavioral assay. Thus for most odorants (except for the rare extreme cases) it could be difficult to assign a clear valence. This would make it makes difficult to establish where absolute valence is encoded in the lateral horn. The authors should consider this issue and discuss potential difficulties in unambiguously defining the “valence” of an odor, as otherwise the literature may be full of contradictory observations*.

We agree with the reviewers that different behavioral assays for testing olfactory preferences in flies sometimes lead to contradictory results. However, 12 out of the 14 odors were also tested in a trap assay ([24], CellReports; Stoekl et al., 2010, CurrBiol; trap assay results for balsamic vinegar are unpublished) and 8 were analyzed using the FlyWalk paradigm ([51], SciReports; pers. comm. Markus Knaden). Notably, all odors, except one, induced the same valence in the trap assay, while 5 odors out of 8 evoked the same behavioral response in the FlyWalk. We added a supplemental figure comparing the odor valence in different behavioral assays (Figure 6—figure supplement 1). In those cases where we observed a difference, the behavioral response was shifted to neutral, i.e. no response was observed. However, we never ever observed that an odor was attractive in one behavioral assay and aversive in another and *vice versa*. We therefore think that the valence that we determined with our behavioral assay in this study seems to be representative for those odors that we have selected. We discussed this point now in our manuscript and added a figure supplement to Figure 6 as referred to above.